# Syndiotactic Polyolefins by Hydrogenation of Highly Stereoregular 1,2 Polydienes: Synthesis and Structural Characterization

**DOI:** 10.3390/polym16192711

**Published:** 2024-09-25

**Authors:** Giovanni Ricci, Ivana Pierro, Antonella Caterina Boccia

**Affiliations:** 1CNR-Istituto di Scienze e Tecnologie Chimiche “Giulio Natta” (SCITEC), Via A. Corti 12, I-20133 Milano, Italy; antonella.boccia@scitec.cnr.it; 2Scientific Advisor, I-21052 Busto Arsizio (VA), Italy; ivanapierro@gmail.com

**Keywords:** stereoregular poly(1,3-diene)s, hydrogenation, syndiotactic polyolefins, polymer characterization, NMR

## Abstract

Some syndiotactic-rich polyolefins, generally difficult to synthesize through stereospecific polymerization of the corresponding monomers, were prepared by homogeneous non-catalytic hydrogenation of syndiotactic 1,2 poly(1,3-diene)s with diimide, arising from thermal decomposition of p-toluene-sulfonyl-hydrazide. All the polymers synthesized were structurally characterized by means of several analytical techniques, such as FT-IR, NMR (^1^H, ^13^C and 2D), DSC, and GPC, and herein illustrated.

## 1. Introduction

In the last years the stereospecific polymerization of 1,3-dienes (i.e., 1,3-butadiene and substituted 1,3-butadienes) [1] has mainly focused on catalytic systems based on the combination of aluminoxanes (e.g., methylaluminoxane (MAO), tetraisobutylaluminoxane (TIBAO)), and different transition metal (e.g., Ti, V, Cr, Co, Fe, Cu) and lanthanide (e.g., Nd, Pr, Gd, La) complexes (e.g., metallocene complexes, phosphine complexes, bypiridyl complexes, bis-imine complexes, imino-pyridine complexes, bis-imino-pyridine complexes) [2,3]. This trend has permitted the synthesis of highly stereoregular poly(1,3-diene)s with different structures (*cis*-1,4; *trans*-1,4; 1,2; 3,4; iso- and syndiotactic) depending on monomer and catalyst structure [4]. In particular, highly syndiotactic 1,2 polymers were obtained with the system CpTiCl_3_/MAO from (Z)-1,3-pentadiene [5,6] and 4-methyl-1,3-pentadiene [6]; highly syndiotactic 1,2 polybutadiene was obtained with chromium (e.g., CrCl_2_ (dmpe)_2_/MAO (dmpe = 1,2-bis (dimethylphosphino)ethane)) [7], cobalt (e.g., (η^3^-C_8_H_13_)(C_4_H_6_)Co) [8] and iron (e.g., FeCl_2_ (bipy)_2_/MAO) [9,10] catalysts; syndiotactic 1,2 polymers were also prepared from terminally substituted 1,3-butadienes (e.g., 1,3-pentadiene, 1,3-hexadiene, 5-methyl-1,3-hexadiene, 1,3-heptadiene, 1,3-octadiene) using the system CoCl_2_ (P*^i^*PrPh_2_)_2_/MAO [11,12,13], while isotactic (E)-1,2 poly(3-methyl-1,3-pentadiene) was synthesized with the system CoCl_2_ (P*^n^*PrPh_2_)_2_/MAO [14]; syndiotactic 3,4 polyisoprene [9,15,16,17] was obtained with iron- and copper-based catalysts (i.e., FeEt_2_ (bipy)_2_-MAO, FeCl_2_ (bipy)_2_/MAO and CuCl_2_ (bipy)/MAO (bipy = bipyridine)); syndiotactic (E)-1,2 poly(3-methyl-1,3-pentadiene) [9,15,18] was instead obtained with the systems FeEt_2_ (bipy)_2_-MAO and FeCl_2_ (bipy)_2_/MAO (bipy = bipyridine).

The results obtained allowed us to improve the knowledge on the diene polymerization mechanism [1,2,3,4], in particular highlighting the strong impact of the monomer structure (i.e., presence of substituents on the monomeric unit) and of the catalyst structure (i.e., nature of the ligand on the metal atom) on the polymerization regio- and stereo-selectivity.

Nevertheless, the extensive accessibility of all these extremely stereoregular polydienes may represent an exceptional and considerable source for highly regular olefin polymers through a simple hydrogenation reaction of the polydienes, as illustrated in Figure 1, especially in the case where such polyolefins cannot be obtained with such a high degree of stereoregularity by simple stereospecific polymerization of the corresponding olefins with transition metal catalysts.

This hydrogenation methodology has been previously applied to prepare highly stereoregular isotactic poly((R,S)-3-methyl-1-pentene) [19,20], and syndiotactic poly(3-methyl-1-butene) [21], that were microstructurally characterized.

In this study, we now report on the preparation and structural characterization by NMR—nuclear magnetic resonance spectroscopy (^1^H, ^13^C and 2D)—of the following syndiotactic-rich polyolefins synthesized by hydrogenation of the diene polymers obtained with the catalytic system CoCl_2_ (P*^i^*PrPh_2_)_2_-MAO (Figure 1): (i) syndiotactic poly(1-pentene) [**H**(**1,2^sy^PP**)] from syndiotactic *trans*-1,2 poly(1,3-pentadiene) (**1,2^sy^PP**) [11]; (ii) syndiotactic poly(1-hexene) [**H**(**1,2^sy^PHX**)] from syndiotactic *trans*-1,2 poly(1,3-hexadiene) (**1,2^sy^PHX**) [11,13]; (iii) syndiotactic poly(5-methyl-1-hexene) [**H**(**1,2^sy^P5MHX**)] from *trans*-1,2 poly(5-methyl-1,3-hexadiene) (**1,2^sy^P5MHX**) [12]; (iv) syndiotactic poly(1-heptene) [**H**(**1,2^sy^PHP**)] from syndiotactic *trans*-1,2 poly(1,3-heptadiene) (**1,2^sy^PHP**) [12]; (v) syndiotactic poly(1-octene) [**H**(**1,2^sy^PO**)] from *trans*-1,2 poly(1,3-octadiene) (**1,2^sy^PO**) [12].

## 2. Materials and Methods

### 2.1. General Procedures and Materials

All the operations were carried out under an inert atmosphere using a dual vacuum/nitrogen line and standard Schlenk-line techniques. Toluene (Sigma-Aldrich, ≥99.7% pure) was refluxed over Na for about 8 h and then distilled and stored over molecular sieves under nitrogen. *o*-Xylene (Sigma-Aldrich, Milano, MI, Italy, pure anhydrous grade), p-toluenesulfonylhydrazide (TSH, Sigma-Aldrich, Milano, MI, Italy), and deuterated solvent for NMR measurements (C_2_D_2_Cl_4_) (Cambridge Isotope Laboratories, Inc., Tewksbury, MA, USA) were used as received. (E)-1,3-pentadiene (Sigma-Aldrich, Milano, MI, Italy, 96% pure), 1,3-hexadiene (Aldrich, Milano, MI, Italy, 99% pure, mixture of (E) and (Z) isomers), 5-methyl-1,3-hexadiene (Chemsampco, Dallas, TX, USA, 80% pure; predominantly E isomer), 1,3-heptadiene (Chemsampco, Dallas, TX, USA, 99% pure; predominantly E isomer), and 1,3-octadiene (Chemsampco, Dallas, TX, USA, 99% pure; predominantly E isomer) were refluxed over calcium hydride for about 4 h, then distilled trap-to-trap and stored under dry nitrogen. Methylaluminoxane (MAO) (10 wt% solution in toluene, Sigma-Aldrich) was used as received. CoCl_2_ (P^i^PrPh_2_)_2_ [11], syndiotactic *tran*s-1,2 poly(1,3-pentadiene) (**1,2^sy^PP**) [11], syndiotactic *trans*-1,2 poly(1,3-hexadiene) (**1,2^sy^PHX**) [11,13], syndiotactic *trans*-1,2 poly(5-methyl-1,3-hexadiene) (**1,2^sy^P5MHX**) [12], syndiotactic *trans*-1,2 poly(1,3-heptadiene) (**1,2^sy^PHP**) [12], and syndiotactic *trans*-1,2 poly(1,3-octadiene) (**1,2^sy^PO**) [12] were prepared as reported in the literature.

### 2.2. Hydrogenation Procedure

The poly(1,3-diene) was introduced into a flask and xylene was subsequently added. The reaction mixture thus obtained was kept under vigorous stirring at room temperature until complete solubilization of the polymer, and then TSH was added. The resulting suspension was refluxed at 120 °C for 3 days, after which it was allowed to spontaneously reach room temperature before adding again the TSH. This operation was repeated once again. Once the reaction was completed, the hydrogenated sample was hot-filtered, the volume of the solution was reduced under vacuum, and methanol was added to coagulate the polymer, which was successively recuperated by filtration. The polymer, once dried under vacuum at room temperature, was extracted with acetone with a Soxhlet for 10 h to remove any excess TSH and byproducts from TSH decomposition. The residual polymer was finally dried under vacuum, dissolved in toluene, precipitated into methanol, and dried again under vacuum at room temperature to constant weight.

The following reaction conditions were applied for the hydrogenation of each polymer:

**1,2^sy^PP***_Syndiotactic trans-1,2 poly(1,3-pentadiene)*: polymer 1 g; xylene, 200 mL; first addition of THS, 9.0 g (5.0 × 10^−2^ mol); second and third addition of THS, 10.0 g (5.56 × 10^−2^ mol). Final yield of **H**(**1,2^sy^PP**)*:* 0.824 g;

**1,2^sy^PHX***_Syndiotactic trans-1,2 poly(1,3-hexadiene)*: polymer 1.9 g; xylene, 300 mL; first addition of THS, 10.0 g (0.54 × 10^−1^ mol); second and third addition of THS, 20.0 g (1.07 × 10^−1^ mol). Final yield of **H**(**1,2^sy^PHX**): 1.64 g;

**1,2^sy^P5MHX***_Syndiotactic trans-1,2 poly(5-methyl-1,3-hexadiene)*: polymer 1.09 g; xylene, 250 mL; first addition of THS, 9.0 g (4.8 × 10^−2^ mol); second and third addition of THS, 11.0 g (5.9 × 10^−2^ mol). Final yield of **H**(**1,2^sy^P5MHX**)*:* 0.89 g;

**1,2^sy^PHP***_Syndiotactic trans-1,2 poly(1,3-heptadiene)*: polymer 0.935 g; xylene, 80 mL; first addition of THS, 6.6 g (3.5 × 10^−2^ mol); second and third addition of THS, 13.0 g (6.9 × 10^−2^ mol). Final yield of **H**(**1,2^sy^PHP**)*:* 0.82 g;

**1,2^sy^PO***_Syndiotactic trans-1,2 poly(1,3-octadiene)*: polymer 1.18 g; xylene, 250 mL; first addition of THS, 9.6 g (5.1 × 10^−2^ mol); second and third addition of THS, 10.0 g (5.9 × 10^−2^ mol). Final yield of **H**(**1,2^sy^PO**)*:* 0.83 g.

### 2.3. Polymer Characterization

Infrared spectroscopy (FTIR) spectra were recorded at room temperature in the 4000–600 cm^−1^ range using a Perkin Elmer Spectrum Two spectrometer (Perkin Elmer Italia S.p.A. 20900 Monza, Italia).

^13^C and ^1^H NMR spectra were recorded on Bruker Avance 400 spectrometer. The spectra were obtained in C_2_D_2_Cl_4_ at 103 °C (hexamethyldisiloxane, HMDS, as internal standard). The concentration of polymer solutions was about 10 wt%. ^13^C parameters were: spectral width 17 kHz; 90° pulse 11.0 μs PL1 −5.0 dB, with a delay of 16 s.

Two-dimensional heteronuclear ^1^H-^13^C experiments were recorded on a Bruker DRX 600 MHz spectrometer, (14.1 T), thermostated at 330 K. The g-HSQC experiment, (gradient-Heteronuclear Single Quantum Correlation), was performed by applying a coupling constant *^1^J_CH_* = 140 Hz; data matrix 2 K × 512; number of scans:128; 7.47 μs as 90° pulse. The g-HMBC experiments (gradient-heteronuclear multiple bond correlation) were performed by applying a delay of 50 ms for the evolution of long-range coupling; data matrix 2 K × 512; number 150 of scans 128; D1 2.00 s. Data were zero-filled and weighted with a sine bell function before Fourier transformation. The microstructure of the resultant polymers was determined by ^1^H and ^13^C NMR, according to the literature data.

The molecular weight averages (*M*_w_) and the molecular weight distribution (*M*_w_/*M*_n_) were determined by a high-temperature Waters GPCV2000 size exclusion chromatography (SEC) system using two online detectors: a differential viscometer and a refractometer. The experimental conditions consisted of three PL Gel Olexis columns, *o*-DCB (*ortho* dichlorobenzene) as the mobile phase, 0.8 mL min^−1^ flow rate, and 145 °C temperature. Universal calibration of the SEC system was performed using eighteen narrow *M*_w_/*M*_n_ polystyrene standards with molar weights ranging from 162 to 5.6 × 10^6^ g mol^−1^. For the analysis, about 12 mg of the polymer was dissolved in 5 mL of o-DCB with 0.05% of BHT as the antioxidant.

## 3. Results and Discussion

Data concerning the preparation of syndiotactic 1,2 poly(1,3-diene)s, such as (**1,2^sy^PP**) [11], (**1,2^sy^PHX**) [11,13], (**1,2^sy^P5MHX**) [12], (**1,2^sy^PHP**) [12] and (**1,2^sy^PO**) [12], are summarized in Table 1.

The non-catalytic hydrogenation with diimide [22,23,24,25,26] of the above-reported polymers herein illustrated (Figure 1) was carried out at 120 °C in *o*-xylene in homogeneous conditions. Practically, the diimide molecule (N_2_H_2_), formed in situ through the thermal decomposition of *p*-toluenesulfonyl hydrazide (TSH), is capable to release the hydrogen molecule responsible of the hydrogenation reaction of the olefinic double bond.

Hereafter it is reported the adopted nomenclature referring to the hydrogenated polymers: [**H**(**1,2^sy^PP**)] is the syndiotactic poly(1-pentene), [**H**(**1,2^sy^PHX**)] is the syndiotactic poly(1-hexene), [**H**(**1,2^sy^P5MHX**)] is the syndiotactic poly(5-methyl-1-hexene), [**H**(**1,2^sy^PHP**)] is the syndiotactic poly(1-heptene)}, and [**H**(**1,2^sy^PO**)] is the syndiotactic poly(1-octene).

Data concerning the syndiotactic degree, which essentially remains that of the starting 1,2 poly(1,3-diene), molecular weight and molecular weight distribution for all the hydrogenated polymers are summarized in Table 2.

### 3.1. Syndiotactic Poly(1-pentene)_H(1,2^sy^PP)

Syndiotactic-rich poly(pentene) was prepared via hydrogenation of *trans*-1,2 poly(1,3-pentadiene), synthesized with the catalyst system CoCl_2_ (P*^i^*PrPh_2_)_2_/MAO as reported in Section 2.2, and illustrated in Figure 2.

NMR data, such as ^1^H and ^13^C NMR spectra (Figure 1), confirmed the completeness of the hydrogenation diene due to the disappearance of the characteristic olefinic signals in the spectra.

Figure 1 shows the comparison of NMR data of syndiotactic *trans*-1,2 poly(1,3-pentadiene) and of the hydrogenated product.

The structure and tacticity of the obtained syndiotactic poly(1-pentene) was analyzed by means of ^1^H and ^13^C NMR and verified according to literature data [27,28,29].

### 3.2. Syndiotactic Poly(1-hexene)_H(1,2^sy^PHX)

Syndiotactic-rich poly(1-hexene) [**H**(**1,2^sy^PHX**)] has been prepared by hydrogenation of syndiotactic *trans*-1,2 poly(1,3-hexadiene), synthesized with the catalyst system CoCl_2_ (P*^i^*PrPh_2_)_2_/MAO, as shown in Figure 3.

The hydrogenation product (i.e., syndiotactic poly(1-hexene)) features are listed in Table 2.

Comparing the ^1^H and ^13^C NMR spectra (Figure 2) of the starting syndiotactic *trans*-1,2 poly(1,3-hexadiene) with those of the corresponding hydrogenated product, it was possible to establish the successful complete hydrogenation reaction.

The structure and tacticity of the resulting syndiotactic poly(1-hexene) was examined through ^1^H and ^13^C NMR, as shown in Appendix A [28,29,30,31].

The microstructure of the hydrogenated polymer was investigated through two-dimensional NMR experiments, such as HSQC (heteronuclear single quantum coherence), which is used to determine the proton–carbon single-bond correlations. As for the polypropylene, the simplest polymer, the syndiotactic poly(1-hexene) obtained after the hydrogenation process of the *trans*-1,2 poly(1,3-hexadiene) has the two protons on the carbon C1 magnetically equivalent, as they experienced the same chemical environment. The NMR equivalence of the C1 protons was confirmed by the presence in the HSQC spectrum (Appendix A) of a single cross peak (at δ_C_ = 39.66 ppm) associated to the single bond correlation of the two protons (δ_C_ = 1.037 ppm) linked to the C1 carbon atom. Practically, the saturated polymer retains the microstructure of the native polymer even after the hydrogenation process. The correctness of the protons and carbons assignment was also confirmed trough the HMBC experiments (Appendix A) which provided correlations between protons and carbons that are two or three bonds away.

### 3.3. Syndiotactic Poly(5-methyl-1-hexene) [H(1,2^sy^P5MHX)]

The syndiotactic-rich poly(5-methyl-1-hexene) was obtained by hydrogenation of the syndiotactic *trans*-1,2 poly(5-methyl-1,3-hexadiene) obtained by polymerizing 5-methyl-1,3-hexadiene with the system CoCl_2_ (P*^i^*PrPh_2_)_2_-MAO (Figure 4).

The hydrogenation product (i.e., syndiotactic poly(5-methyl-1-hexene) features are listed in Table 2.

The completion of the hydrogenation reaction was confirmed by comparing of the FT-IR spectra (Appendix A) and the NMR (^1^H and ^13^C; Figure 3) spectra of the starting syndiotactic *trans*-1,2 poly(1,3-diene) to those of the corresponding hydrogenated product.

In the FT-IR spectrum of the syndiotactic poly(5-methyl-1-hexene) (Appendix A), the strong band at 965 cm^−1^ observed in the spectrum of *trans*-1,2 poly(5-methyl-1,3-hexadiene), indicative of the presence of a *trans* double bond, is completely absent.

The structure and tacticity of the resulting syndiotactic 1,2 poly(5-methyl-1-hexene) were determined through NMR (^1^H and ^13^C). Figure 3 shows the ^13^C NMR spectra of the syndiotactic *trans*-1,2 poly(5-methyl-1,3-hexadiene) and its saturated polymer together with the peak attribution.

### 3.4. Syndiotactic Poly(1-heptene)_H(1,2^sy^PHP)

Syndiotactic-rich poly(1-heptene) was obtained by hydrogenation of syndiotactic *trans*-1,2 poly(1,3-heptadiene) synthesized by polymerizing 1,3-heptadiene with the system CoCl_2_ (P*^i^*PrPh_2_)_2_-MAO (Figure 5).

The hydrogenation product (i.e., syndiotactic poly(1-heptene) features are listed in Table 2.

The complete hydrogenation of the diene polymer was evident by comparing the ^1^H NMR spectra (Figure 4) of the starting syndiotactic *trans*-1,2 poly(1,3-heptadiene) with the spectra of the resulting hydrogenated product. The signals in the olefinic region (from 5.2 to 5.4 ppm), detected in the ^1^H NMR spectrum of the diene polymer and due to the olefinic hydrogen atoms, are not observed in the ^1^H NMR spectrum of the hydrogenated polymer, according indeed with the complete hydrogenation of the diene polymer.

The structure and tacticity of the resulting syndiotactic poly(heptene) were examined by ^1^H and ^13^C NMR [28,31,32]. Figure 4 shows the ^13^C NMR spectra of the syndiotactic *trans*-1,2 poly(heptadiene) and its saturated polymer together with the peak attribution.

The microstructure of the hydrogenated syndiotactic poly(1-heptene) was investigated through two-dimensional HSQC experiments (Appendix A), observing that the two protons on the carbon C1 were magnetically equivalent meaning they experienced the same chemical environment. The NMR equivalence of the C1 protons was confirmed by the presence in the HSQC spectrum of a single cross peak associated to the single bond correlation of the two protons (at δ_C_ = 1.02 ppm) linked to the C1 carbon atom (δ_C_ = 39.72 ppm). As for the H(1,2syPHX) polymer, the correctness of the protons and carbons assignment was confirmed with the HMBC experiments (Appendix A).

### 3.5. Syndiotactic Poly(1-octene)_[H(1,2^sy^PO)]

Syndiotactic *trans*-1,2 poly(1,3-octadiene), obtained by polymerizing 1,3-octadiene with CoCl_2_ (P*^i^*PrPh_2_)_2_-MAO, was treated with tosylhydrazide providing syndiotactic poly(1-octene) (Figure 6).

The hydrogenation product (i.e., syndiotactic poly(1-octene) features are listed in Table 2.

The comparison between the two ^1^H NMR spectra (Figure 5) of the starting syndiotactic *trans*-1,2 poly(1,3-diene) and of the resulting hydrogenated product highlights the complete hydrogenation of the starting polydiene.

In the FT-IR spectrum of the syndiotactic poly(octene) (Appendix A), the strong band at 967 cm^−1^ indicative of the presence of a *trans* double bond is not detectable, while an intense band at 735 cm^−1^, attributed to the vibration of a –CH_2_– unit, typical of saturated polyolefins, was observed.

The peaks in the olefinic region (from 5.2 to 5.4 ppm), detected in the ^1^H NMR spectrum (Figure 5) of the diene polymer and due to the olefinic hydrogen atoms, are not visible in the ^1^H NMR spectrum of the hydrogenated polymer, confirming that the hydrogenation reaction was completed. The structure and tacticity of the syndiotactic poly(octene) has been examined through of ^1^H and ^13^C NMR [28,31]. Figure 5 shows the ^13^C NMR spectra of the syndiotactic *trans*-1,2 poly(octadiene) and its saturated polymer together with the peak attribution.

## 4. Conclusions

Several syndiotactic-rich polyolefins were prepared by hydrogenation of highly stereoregular syndiotactic 1,2 poly(1,3-diene)s, and their microstructure was determined through NMR analysis. Such highly stereoregular polyolefins could be quite useful as polymer models for the microstructural characterization of analogous polymers, even with lower stereoregularity, which can be obtained by stereospecific polymerization of the corresponding monomers.

The polyolefins obtained could be of potential interest for applications in the elastomeric and/or thermoplastic fields, although it is difficult to imagine an industrial application, given the current high cost of the monomers used for the preparation of the polydienes, from which polyolefins are obtained through hydrogenation. However, in the near future, these niche polymers may be produced at more accessible costs opening the scenario on appealing industry applications.

## Data Availability

The original contributions presented in the study are included in the article/Appendix A, further inquiries can be directed to the corresponding author.

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
