# Peer review of "Syndiotactic Polyolefins by Hydrogenation of Highly Stereoregular 1,2 Polydienes: Synthesis and Structural Characterization"

_polymers, 2024, doi:10.3390/polym16192711_

Round 1
Reviewer 1 Report
Comments and Suggestions for Authors
This paper reports synthesis and structural characterization of syndiotactic polyolefins by hydrogenation of highly stereoregular 1,2 polydienes. This work provides structural information of various syndiotactic polyolefins, especially 13C NMR. I would recommend its publication with revisions.
1. Syndiotactic polyolefins may be not suitable. In the text, the highest [rrrr] value is 62%. Syndiotactic-rich polyolefins may be better. In fact, syndiotactic polyolefins cannot show melting temperature.
2. For 13 NMR spectra, authors only give several main peaks. Some minor peaks should be other structure such as [rmrr] et al. I know that [rrrr] values are calculated based on 1,2 polydienes. Do you give [rrrr] calculation of syndiotactic polyolefins based their 13C NMR?
3. For 1,2syPHP and 1,2syPO, low 1,2-selectivity will make 13C NMR more complex. if possible, author make a discussion.
4. DSC curves should be supplied in SI.
Author Response
REPLIES TO REFEREES' COMMENTS
Reviewer 1
This paper reports synthesis and structural characterization of syndiotactic polyolefins by hydrogenation of highly stereoregular 1,2 polydienes. This work provides structural information of various syndiotactic polyolefins, especially 13C NMR. I would recommend its publication with revisions.
- Syndiotactic polyolefins may be not suitable. In the text, the highest [rrrr] value is 62%. Syndiotactic-rich polyolefins may be better. In fact, syndiotactic polyolefins cannot show melting temperature.
According to the Reviewer suggestion we modified the text adopting the syndiotactic-rich polyolefins definition”, as the syndiotactic degree is not very high, but it is the one deriving from the starting polydienes, since hydrogenation does not modify their chirality. On the other hand, the polydienes used for hydrogenation were obtained with the CoCl2(PiPrPh2)2/MAO system which proved to be the best one in terms of catalytic activity and selectivity.
- For 13 NMR spectra, authors only give several main peaks. Some minor peaks should be other structure such as [rmrr] et al. I know that [rrrr] values are calculated based on 1,2 polydienes. Do you give [rrrr] calculation of syndiotactic polyolefins based their 13C NMR?
The syndiotactic degree exhibited by the 1,2 poly(1,3-diene)s is maintained in the polyolefins resulting from their hydrogenation, since the hydrogenation reaction does not modify the chirality. The degree of syndiotacticity was however evaluated and confirmed by examining the 13C NMR spectra of the polyolefins, specifically considering the integral ratio of selected signals. Furthermore, we added in the SI a 13C NMR spectrum showing the assignment at triad level and how determine the tacticity in %.
- For 1,2syPHP and 1,2syPO, low 1,2-selectivity will make 13C NMR more complex. if possible, author make a discussion.
The referee is right, the 1,2 content in 1,2syPHP and 1,2syPO is lower than in the other polydienes and therefore some other peaks originating from the hydrogenation of random the cis-1,4 units randomly distributed along the polymer chain can be observed, however the sequences of 1,2 units are long enough to be able to evaluate their level of syndiotacticity in a way that is acceptable for our purposes.
- DSC curves should be supplied in SI.
We have not reported the DSC curves because no endothermal event indicative of a melting point was observed, and the Tg values are shown in Table 2.
Reviewer 2 Report
Comments and Suggestions for Authors
The synthesis of syndiotactic polyolefins via the stoichiometric hydrogenation of the corresponding 1,2-polyolefins is described is this manuscript. Since the synthesis of the polyolefin precursors has been already reported by the same research group the current work seems rather weak in terms of the quantity of the presented results. There is no doubt that the performed hydrogenation reaction has been accomplished and the desired products have been prepared. However, the authors have to respond to the following suggestions and criticisms before a possible publication of this work:
· The authors have to stress why this work is important. Maybe, a possible industrial application is not possible. However, what kind of scientific questions can be answered using these products?
· Why did the authors choose to perform a stoichiometric hydrogenation reaction using TSH, instead of a catalytic reaction? What are the advantages of this proposed approach?
· The authors should explain why TSH was not added once at the beginning of the hydrogenation reaction. Instead, it was added in two or three separate doses.
· More data on these samples are needed to strengthen the impact of this work. Emphasis should be given on the thermal properties of these materials (e.g. thermal stability), their mechanical properties, their flexibility etc.
· Almost all the references in the manuscript are papers of the authors. I think that there is a limit of self-citations from the journal.
Comments on the Quality of English LanguageThere are only a few minor points to correct. The overall presentation is very good.
Author Response
Reviewer 2
The synthesis of syndiotactic polyolefins via the stoichiometric hydrogenation of the corresponding 1,2-polyolefins is described is this manuscript. Since the synthesis of the polyolefin precursors has been already reported by the same research group the current work seems rather weak in terms of the quantity of the presented results. There is no doubt that the performed hydrogenation reaction has been accomplished and the desired products have been prepared. However, the authors have to respond to the following suggestions and criticisms before a possible publication of this work:
- The authors have to stress why this work is important. Maybe, a possible industrial application is not possible. However, what kind of scientific questions can be answered using these products?
As reported in the text, the dienes used are currently extremely expensive and therefore unlikely to find industrial application. However, for instance, in recent times pentadiene is attracting some interest (cis and trans polymers have shown interesting elastomeric properties), and pentadiene represents a significant part of the C5 fraction which primarily derives from the cracking process, together with isoprene and cyclopentadiene, so it cannot be excluded that in the near future the dienes in question may be produced at more accessible costs. At the moment, as we have reported in the text, we believe that the syndiotactic polyolefins obtained by hydrogenation may be useful as models for the characterization of similar polymers obtained through stereospecific polymerization of the corresponding olefins.
- Why did the authors choose to perform a stoichiometric hydrogenation reaction using TSH, instead of a catalytic reaction? What are the advantages of this proposed approach?
The hydrogenation of dienes with TSH has the advantage of being carried out under safer experimental conditions than those involving the hydrogen (e.g. high pressure reactors, etc).
- The authors should explain why TSH was not added once at the beginning of the hydrogenation reaction. Instead, it was added in two or three separate doses.
TSH is used in large excess to achieve complete hydrogenation, and we have experimentally verified that by adding tosylhydrazide in a single solution the reaction is too violent; we therefore opted for the three subsequent additions for safely carrying out of the reaction.
- More data on these samples are needed to strengthen the impact of this work. Emphasis should be given on the thermal properties of these materials (e.g. thermal stability), their mechanical properties, their flexibility etc.
We have carried out some analyses in order to evaluate the mechanical properties of these materials but the results were not comforting and therefore we have limited the report to the structural characterization. However, we are still working on this aspect.
- Almost all the references in the manuscript are papers of the authors. I think that there is a limit of self-citations from the journal.
At the publisher's suggestion, we have already removed several citations of our papers, but we cannot do much more as the literature is not richer of data on these niche materials. We could only cite the 2021 review on Macromolecules, which contains all the necessary references, but I think that it would not be very fair towards the other authors.
Reviewer 3 Report
Comments and Suggestions for Authors
A series of syndiotactic polyolefins were prepared in two steps (i) by polymerization of 1,3-dienes leading to syndiotactic poly1,2(1,3 dienes) and (ii) by following homogeneous non-catalytic hydrogenation. This method was described earlier for model monomers and now it was applied for new ones. It was found by IR and NMR technique that hydrogenation was quantitative and tacticity was preserved. I recommend accepting this manuscript for publication after a minor revision.
1. In Introduction the author should explain shortly what are the advantages of this two- step synthesis (from the point of view of monomer availability, final polymer tacticity and applicability).
2. Table 2 shows preservation of several important polymer characteristics during hydrogenation. However, in the case of poly (5-methyl 1-hexene) a strong decrease of polydispersity index (from 4.3 to 1.9) was observed. Could you comment it, please?
3. As polymer tacticity is a key parameter, I recommend to illustrate its determination (for example as a Figure in Supporting material).
4. The discussion of HMBC spectra (Fig. S3 and S4) should be added into the text on proper places (line 232 and 300).
5. Conclusions paragraph is rather poor. The majority of it belongs to Introduction. Please improve this paragraph.
6. The abbreviation o-DCB should be explained.
7. The amount of self-citation seems to me high (56% of all references).
Author Response
Reviewer 3
A series of syndiotactic polyolefins were prepared in two steps (i) by polymerization of 1,3-dienes leading to syndiotactic poly1,2(1,3 dienes) and (ii) by following homogeneous non-catalytic hydrogenation. This method was described earlier for model monomers and now it was applied for new ones. It was found by IR and NMR technique that hydrogenation was quantitative and tacticity was preserved. I recommend accepting this manuscript for publication after a minor revision.
- In Introduction the author should explain shortly what are the advantages of this two- step synthesis (from the point of view of monomer availability, final polymer tacticity and applicability).
In order to comply with the referee's request, we have modified part of the introduction as follows: “Nevertheless, the extensive accessibility of all these extremely stereoregular polydienes may represent an exceptional and considerable source for highly regular olefin polymers through a simple hydrogenation reaction of the polydienes, as illustrated in Scheme 1, especially in the case where such polyolefins cannot be obtained with such a high degree of stereoregularity by simple stereospecific polymerization of the corresponding olefins with transition metal catalysts.
- Table 2 shows preservation of several important polymer characteristics during hydrogenation. However, in the case of poly (5-methyl 1-hexene) a strong decrease of polydispersity index (from 4.3 to 1.9) was observed. Could you comment it, please?
The referee is absolutely right. The correct value of the polydispersity index is 2.9 instead of 1.9, as erroneously reported in the text. This does not take away from the fact that the decrease is evident: the explanation I feel like giving is that in the hydrogenation process a certain amount of low molecular weight polymer was lost.
- As polymer tacticity is a key parameter, I recommend to illustrate its determination (for example as a Figure in Supporting material).
Ok done, and thanks for the suggestion, we added in the SI a 13C NMR spectrum showing the assignment at triad level and how determine the tacticity in %.
- The discussion of HMBC spectra (Fig. S3 and S4) should be added into the text on proper places (line 232 and 300).
We added a comment into the text, thank you.
- Conclusions paragraph is rather poor. The majority of it belongs to Introduction. Please improve this paragraph.
We have modified the conclusions paragraph as follows, hoping that this will satisfy the referee's request: “Several syndiotactic polyolefins were prepared by hydrogenation of highly stereoregular syndiotactic 1,2 poly(1,3-diene)s, and their microstructure was determined through NMR analysis . Such highly stereoregular polyolefins could be quite useful as polymer models for the microstructural characterization of analogous polymers, even with lower stereoregularity, which can be obtained by stereospecific polymerization of the corresponding monomers.
The polyolefins obtained could be of potential interest for applications in the elastomeric and/or thermoplastic fields, although it is difficult to imagine an industrial application, given the current high cost of the monomers used for the preparation of the polydienes from which polyolefins are obtained through hydrogenation. However, in the near future these niche polymers may be produced at more accessible costs opening the scenario on appealing industry applications.
- The abbreviation o-DCB should be explained.
Ok done.
- The amount of self-citation seems to me high (56% of all references).
At the publisher's suggestion, we have already removed several citations of our papers, but we cannot do much more as the literature is not richer of data on these niche materials. We could only cite the 2021 review on Macromolecules, which contains all the necessary references, but I think that it would not be very fair towards the other authors.
Round 2
Reviewer 2 Report
Comments and Suggestions for Authors
The scientific part of this work is sufficient and very interesting to the polymers community. I will expect further elucidation of this research describing the properties of these materials. For the moment this manuscript can be published in Polymers in its current form.
Author Response
Thanks to the referee for the positive comment and the suggestion.
We are already working to evaluate the properties of these materials.